# Preimplantation Genetic Testing for Aneuploidy (PGT-A) in In-Vitro Fertilisation (IVF) Treatment: Study Protocol for Pilot Phase of a Randomised Controlled Trial

**DOI:** 10.3390/jcm13206192

**Published:** 2024-10-17

**Authors:** Yusuf Beebeejaun, Kypros H. Nicolaides, Anastasia Mania, Ippokratis Sarris, Sesh K. Sunkara

**Affiliations:** 1King’s Fertility, Fetal Medicine Research Institute, King’s College Hospital, London SE5 8BB, UK; 2Department of Women’s and Children’s Health, Faculty of Life Sciences and Medicine, King’s College London, London SE1 7EH, UK; 3Harris Birthright Research Centre of Fetal Medicine, Fetal Medicine Research Institute, King’s College Hospital, London SE5 8BB, UK

**Keywords:** pre-implantation genetic testing, aneuploidy, embryo mosaicism, IVF

## Abstract

**Introduction**: Poor outcomes following IVF treatments are speculated to be due to the transfer of aneuploid embryos that cannot be identified based on morphological evaluation alone. This leads to patients requiring numerous embryo transfers and, consequently, a prolonged time interval before live birth. Embryo selection following preimplantation genetic testing for aneuploidy (PGT-A) with next-generation sequencing (NGS) has been suggested as an intervention to shorten time to pregnancy in women undergoing in vitro fertilisation (IVF). Past studies assessing the clinical efficacy of PGT-A in improving clinical outcomes have been conflicting and the associated clinical pregnancy rates and live birth rates following the transfer of a mosaic embryos have yet to be determined. None of the existing studies solely included women of advanced reproductive age (ARA). The pilot study and proposed RCT will determine if, compared to morphological evaluation alone, the use of PGT-A through NGS is a more clinically effective, safer, and more cost-effective way to provide IVF treatment in women of advanced reproductive age. **Method and Analysis**: The proposed pilot study will aim to randomise 100 patients within a single-centre study to evaluate recruitment, randomisation, and adherence to study protocol and allocated trail arms by participating patients. The results of the pilot study will enable us to determine the sample size for a larger study to establish the effectiveness of PGT-A in ARA women. **Ethics and Dissemination**: The study (Integrated Research Application System Number 236067) received approval from the Health Research Authority and Health and Care Research Wales (HCRW) and the East Midlands—Leicester South Research Ethics Committee (20/EM/0290). The results will be made available to patients, the funders, the Reproductive Medicine societies, and other researchers. **Trial registration**: ClinicalTrials.gov Identifier: NCT05009745, n.

## 1. Introduction

### 1.1. Background

The selection of the most viable embryo is typically based solely on light microscopy and morphological assessment in assisted reproductive technologies (ART) [1,2,3]. However, despite significant technological and scientific advances in the field, many women still fail to achieve clinical pregnancy or live birth, despite the transfer of embryos classified as good quality by various guidelines [4,5,6,7,8]. This high rate of failure is believed to due to the limitations of relying solely on morphological evaluation, which is an inefficient method of determining embryo ploidy, unable to accurately distinguish between euploid, aneuploid, or mosaic embryos, which have distinct implantation potentials [3,7,8]. The human embryo aneuploidy rate is reported to range between 30% and 50% when under 35 years of age and to increase to 80% in patients aged 42 years of age or older [9,10], reflecting the higher miscarriage rates and lower live birth rates often seen in women of advanced reproductive age. Often, to increase the chances of a live birth, assisted-conception units rely on the transfer of multiple embryos graded as good on morphological criteria alone. However, multiple-embryo transfer is known to increase the risk of multiple pregnancies, which are associated with significant foetal, maternal, and neonatal risks, as well as long-term complications [11]. In view of this, providing patients with an added level of single-embryo selection beyond a static or dynamic morphologic evaluation is of increasing clinical importance.

Preimplantation genetic testing for aneuploidies (PGT-A), combined with next-generation sequencing (NGS), aims to identify and transfer euploid embryos with the highest implantation potential, while excluding aneuploid embryos that are likely to arrest in development and lead to negative pregnancy outcomes [12,13,14]. In additional to shortening the time-to-pregnancy timeline, reducing the miscarriage rate, and improving live birth rates, IVF with PGT-A also aims to promote single-embryo transfer, thereby reducing the risk of multiple births without lowering overall IVF success rates [13,15].

However, with the evolution of PGT-A techniques from fluorescence in situ hybridization (FISH), which only analysed up to eight chromosomes, to NGS, which provides a complete analysis of all 23 chromosomes [16], a limited number of clinically relevant randomised controlled trials are available, and the use of PGT-A, and particularly whether it truly improves live birth rates, remains contentious [17,18]. Earlier studies on FISH are now considered outdated and cannot be applied to current practice [9,19,20]. Another concern with past technology was the inherent risk of inaccuracies due to signal, which increases the rate of false negatives and false positives. Past technology also lacked the ability to define, detect, and differentiate between embryos with low and high mosaic genetic constitutions, a common finding in IVF-derived human embryos [21].

### 1.2. Current Evidence

Currently, few trials exist assessing the clinical benefit of IVF with PGT-A via NGS vs. IVF with no PGT-A, and a meta-analysis of their findings were inconclusive. Challengers of PGT-A dismiss earlier trials due to concerns over the ‘intention-to-treat’ criteria, focusing instead on the ESTEEM and STAR trials, which indicated that PGT-A is effective in increasing pregnancy rates [19,22]. However, supporters of PGT-A argue that these trials, along with all RCTs conducted so far, demonstrate at least some positive outcomes for PGT-A, arguing that participants in the STAR trial had an average age of 28 years old, which is associated with a low aneuploid rate. However, post hoc analysis in women aged 35–40 suggested a notable increase in OPR per embryo transfer for PGT-A cycles [19]. Challengers argue that this approach is not valid, despite the results of both the STAR trial and the SART study, which compiled data from U.S. IVF clinics, both showing statistically significant differences in the over-35 age category [19,23]. These findings were described as contentious and not deemed reliable by the HFEA’s Scientific and Clinical Advances Advisory Committee (SCAAC). Other studies have faced significant methodological criticisms, suggesting they contained underpowered study designs, selection bias, inadequate blinding, and a lack of multi-centre validation.

Superficially, even the latest multicentre study detailing clinical outcomes from 1212 patients did not identify IVF without PGT-A as being associated with superior outcomes compared to PGT-A [24]. However, this study has been criticised for its flaws—it had an average participant age of 29 years old, a euploidy rate of 69.8%, and the methodology used appear to be biassed against PGT-A, with the use of cumulative live birth rate as a primary outcome [23], which may not accurately reflect PGT-A’s superiority due to these biases and differences in embryo selection and transfer practices. The authors also only selected three top-quality embryos, based on morphology, for biopsy and did not perform a biopsy on all available embryos, which is routine clinical practice. The study also did not account for the transfer of low-grade mosaic embryos, which can be performed in real life clinical care [24]. Therefore, the clinical effectiveness of PGT-A as a selection tool in women with an expected high aneuploidy rate, namely those above the age of 35 years, remains to be determined.

Our study’s novelty lies in its focus on an older age group, where the prevalence of aneuploidy and mosaicism is higher, and evaluation of the impact of advanced PGT-A methods on live birth rates. Additionally, we will specifically address the outcome of mosaic embryos—a critical aspect that was often overlooked in previous studies. By examining how mosaic embryos affect clinical outcomes and incorporating them into the analysis, our study will provide valuable insights into the practical implications of the use of PGT-A for diverse embryo classifications.

By addressing these aspects, we aim to offer new evidence on the effectiveness of PGT-A in improving IVF outcomes, particularly for older women and in the context of mosaic embryo management.

## 2. Support for This Study

The need for a robust RCT assessing the potential clinical benefit of PGT-A in women of advanced reproductive age is recognised by the European Society of Human Reproduction (ESHRE), which has also recognised that the cumulative live birth rate (CLBR) may not effectively demonstrate PGT-A’s superiority due to selection bias, the exclusion of potentially viable embryos, and differences in multiple transfer opportunities [25].

With advancements in embryo genetic screening, there is now a reinvigorated interest in the use of the new PGT-A with NGS in IVF to reduce miscarriage rates and time to pregnancy, especially in women of advanced reproductive age; this is listed as a research recommendation by the European Society of Human Reproduction [21,25].

In the context of this pilot study, our design will facilitate the evaluation of embryo ranking based on implantation potential, including the assessment of mosaic embryos. However, given the exploratory nature of the pilot, our findings will be preliminary and focused on feasibility. We aim to gather initial data that will inform future studies, rather than to draw definitive conclusions about clinical pregnancy rates at this stage.

### 2.1. Objective

In the proposed pilot study, we will evaluate recruitment, randomisation, and adherence to study protocol and allocated trail arms by participating women. We aim to incorporate the results of the pilot study into the main RCT, for which ethical approval will be sought. Participating women will be informed of our intention to include the pilot data in the full study.

The purpose of the RCT will be to determine if a policy of embryo selection and transfer based on morphological evaluation and PGT-A with NGS is a more clinically effective, safer, more cost-effective, and acceptable way to provide IVF treatment in women of advanced reproductive age compared to the routine practice of embryo selection and transfer based on morphological evaluation alone.

### 2.2. Trial Design

The study will be a prospective, allocation concealed, two-arm, parallel RCT. Trained research personnel will recruit eligible patients from within the assisted-conception clinic and obtain informed consent using the agreed consent form.

### 2.3. Study Setting and Timeline

The pilot phase of the study, to be conducted at King’s Fertility within King’s College Hospital NHS Foundation Trust, will provide an estimate of the recruitment rate, randomisation rate, and adherence to the study.

### 2.4. Inclusion Criteria

Any women aged 35–42 years undergoing IVF ± ICSI treatment for infertility, with the aim of a fresh embryo transfer, will be eligible to participate in the study.

The trial population was chosen to be consistent with expert (national and international) consensus opinion and a UK-based survey of fertility specialists which suggested that women of advanced reproductive age (35 years and above) undergoing IVF are most likely to benefit from PGT-A.

### 2.5. Exclusion Criteria

Patients undergoing preimplantation genetic testing for inherited genetic disorders;Gamete donation cycles;Patients with untreated hydro-salpinges;Patients with untreated uterine pathology (e.g., endometrial polyps, submucous fibroids, intramural fibroids > 5 centimetres in maximum diameter, intrauterine adhesions, uterine septa);Patients currently participating in research involving interventions or who were recently (within a month) involved in interventional research.

### 2.6. Consent

Women aged between 35 years and 42 years old who meet the inclusion and exclusion criteria and who wish to undertake an IVF/ICSI treatment cycle will be invited to participate in the trial. Recruitment will continue through monitoring scans up to the day of egg collection, and eligible women and patients will be given a participant information sheet about the trial. If a woman and her partner agree to participate in the trial, a formal signed trial consent) will be obtained from both partners, conducted in accordance with ethical standards.

### 2.7. Randomisation

#### Software

Third party, distant, internet-based block randomisation with minimisation will be used to ensure randomisation and complete allocation concealment. Women will be randomised into the trial using a secure online randomisation system, accessible via the MedSciNet Clinical Trial Framework (www.medscinet.net). This system will be available 24/7, with a toll-free telephone randomisation service available from Monday to Friday, 09:00 to 17:00 UK time, excluding bank holidays and university closure days. This option will provide flexibility for researchers needing assistance.

Before randomisation, investigators will complete registration paperwork to collate necessary participant information. All required data must be submitted; if any information is missing, randomisation will be paused until it is provided. Upon verification of eligibility and baseline data, the trial number and treatment pack number will be assigned.

### 2.8. Minimisation Algorithm

The randomisation system (allocation ratio 1:1) will employ a minimisation algorithm to ensure balance across the following factors: IVF clinic, female age (35–37, 38–39 and 40–42 years), BMI (20–<25, 25–<30, ≥30), previous live birth, type of ovarian stimulation protocol (gonadotrophin-releasing hormone (GnRH) agonist or GnRH antagonist; human chorionic gonadotrophin (hCG) or GnRH agonist trigger), and standard IVF or ICSI.

### 2.9. Confidentiality and Concealment

Each user will receive unique login credentials, which will be distributed according to the study’s Signature and Delegation Log to ensure proper access control. The randomisation and allocation process will be fully automated to prevent any researcher involvement in participant assignment. The allocation sequence will be concealed and only revealed to the clinical team at the time of participant assignment.

## 3. Blinding

Although it will not be possible to blind the allocation to study arms due to the nature of the PGT-A strategy and the regulatory requirements of the Human Fertilisation and Embryology Act 1990 and the Human Fertilisation and Embryology Code of Practice 2009—which mandate that couples are informed about the fate of their embryos (whether subjected to PGT-A or not)—we will implement several strategies to minimise potential bias from both researchers and participants.

Data analysts will be blinded to participant allocation, ensuring that the interpretation of results remains impartial and unaffected by knowledge of group assignments. All study procedures, including data collection and outcome assessment, will be standardised to minimise variability and reduce bias. Comprehensive training will be provided to all staff to ensure the consistent application of protocols across all study sites.

Regular audits and reviews will be conducted to ensure adherence to protocols and to identify and address any sources of bias. Wherever feasible, outcome assessors will be blinded to participant allocation to prevent their knowledge of the group assignments from influencing the evaluation of the outcomes. Robust measures will be implemented to secure participant data and maintain confidentiality, with access to sensitive information restricted to authorised personnel only, further reducing the risk of bias.

## 4. Primary End Point

Our primary outcome of interest will be the ability to recruit, randomise, and ensure adherence to the study protocol to plan the full study.

## 5. Secondary End Points

The following secondary outcomes will be recorded:Clinical pregnancy rate per transferred embryo.Clinical pregnancy rate per randomised woman.Miscarriage rate per transferred embryo.Time to pregnancy leading to live birth.

## 6. Study Design

### 6.1. Intervention Regimens

The study design describes a single-centre, randomised, placebo-controlled, two-arm study (see Figure 1). The experimental (intervention) strategy will be to perform PGT-A via NGS on good-quality embryos at the fully expanded blastocyst stage.

### 6.2. Timing of Randomisation

Randomisation will occur on day 3 following egg collection. At this point, all consented patients will be evaluated for eligibility. Specifically, patients will be randomised if they have at least three good-quality embryos, as assessed using the embryo-scoring system of the Association of Clinical Embryologists (UK).

### 6.3. Care Pathway Post-Randomisation

Women in the control arm will proceed with a fresh embryo transfer procedure on day 5 following egg collection, or, if clinically indicated, a frozen–thawed embryo transfer as first-line treatment over a fresh transfer. Women in the experimental arm will have PGT-A performed from trophectoderm biopsy on embryos that reach the blastocyst stage, which typically occurs on days 5, 6, or 7 following egg collection.

All embryos will be frozen after the biopsy procedure. The CCS of all chromosomes will be performed using NGS and embryos will be categorised as euploid, mosaic aneuploid, and aneuploid embryos. Mosaic embryos will be ranked based on the degree of aneuploidy (low- and high-mosaic aneuploid embryos).

A low-mosaic embryo is defined as an embryo with whole-chromosome aneuploidies in 30–50% of the biopsied cell and a high-mosaic embryo is defined as an embryo with whole-chromosome aneuploidies in 50–70% of the biopsied cells.

### 6.4. Embryo Transfer

Only single-euploid embryos or low-mosaic embryos that are deemed suitable to be replaced, with prioritisation based on the morphological grading of the embryos, will be transferred in subsequent frozen–thawed embryo transfer cycles, which could be initiated as early as with the next menstrual cycle following egg collection. Euploid embryos will preferably be transferred to low-mosaic embryos and the couple will need to undergo genetic counselling before transferring a low-mosaic embryo.

The subsequent frozen–thawed embryo transfer (FET) of a single-euploid embryo will occur either in a natural cycle or a hormonally medicated cycle using physiological doses of oestrogen and progestogens to prepare the endometrium.

If patients decide not to transfer mosaic embryos, these instances will be thoroughly documented, including the reasons and clinical factors involved. These data will be incorporated into the study through descriptive analysis, which will outline the frequency of and reasons for these decisions. Additionally, sensitivity analyses will assess how excluding mosaic embryos affects the study outcomes, ensuring that the results accurately reflect both the protocol and real-world practices. The final report will address the implications of transferring versus not transferring mosaic embryos on the study’s primary and secondary endpoints, providing a comprehensive understanding of their impact.

### 6.5. Pregnancy Outcome

A pregnancy test will be undertaken around 2 weeks after the embryo transfer in all women. If the pregnancy test is negative, this will be the endpoint for the pilot study for these participants. If the pregnancy test is positive, women will routinely undergo an ultrasound scan after a further 3 weeks to identify the presence of clinical pregnancy, which will be the endpoint for the pilot study for these participants.

Women are requested to report the outcome of the pregnancy, which is a routine practice by all UK clinics, as it is a mandatory requirement for clinics to report IVF treatment outcomes to the Human Fertilisation and Embryology Authority (HFEA), which is the statutory regulator for all assisted-reproduction treatments performed in the UK. Women will be contacted to follow-up the outcomes if they fail to contact the IVF clinic.

Participants will be informed of this and consent to being followed up and to the use of the data from the pilot phase if the study progresses to the full stage. Data on all baseline characteristics and outcomes will be collected using a standardised electronic case report form (eCRF).

### 6.6. Adverse Events

As the trial does not involve any new treatment protocols other than the routine standard care protocols, no serious adverse events are expected. However, all participants will receive information on whom to contact in case of a clinical emergency.

Any potential serious adverse events are likely to be limited to the embryos and may include:Failure of embryos to survive biopsy procedure.Physical integrity of the biopsy samples being compromised in transit.

### 6.7. Statistical Considerations

#### Sample Size

For this pilot feasibility randomised controlled trial, a sample size of 100 women was determined. Patients will be recruited to the study at their initial consultation at the fertility clinic, and further recruitment will continue through their monitoring scans until the day of egg collection. Additional centres may be involved in the full study if it goes ahead. This sample size allows us to estimate recruitment rates with a margin of error of ±10.5% around the true rate, providing an 80% confidence interval. While formal hypothesis testing is not the primary focus of this pilot study, an alpha value of 0.05 (5%) and a beta value corresponding to 80% power (β = 0.2) would be considered standard if a preliminary hypothesis testing were conducted [26]. Given the strict regulatory framework governing IVF in the UK, particularly the mandatory reporting of all IVF treatment outcomes, as required by the Human Fertilisation and Embryology Authority (HFEA), we anticipate nearly complete follow-up, approaching 100%.

### 6.8. Statistical Analysis

As this is a pilot study, the main emphasis will be on the acceptability of the intervention and the randomisation process to couples, and on adherence to the protocol. However, following the “All Trials” principle, we will also present a full analysis appropriate for a randomised controlled trial [27].

Baseline comparisons will be made of female age, previous live birth, cause of infertility, number of previous IVF attempts by the woman, ovarian reserve (assessed by the total antral follicle count: 7 or less; 8–20; more than 20), type of ovarian stimulation protocol (gonadotrophin releasing hormone (GnRH) agonist or GnRH antagonist; human chorionic gonadotrophin (hCG) or GnRH agonist trigger), and standard IVF or ICSI. The results will be presented as percentages or mean (SD), as appropriate.

### 6.9. Interim Data Analysis and Data Monitoring

#### Recruitment Progress

The target of recruiting 60 women out of 100 within 18 months was set as a feasibility threshold. If at least 60 out of 100 participants are recruited within the 18-month period, the study will be deemed to have met the recruitment feasibility threshold, and the study will proceed to the main phase. The investigation will stop without proceeding to the main study if the pilot fails to show adequate recruitment after 18 months. If this target is not met, it will prompt a reassessment of the recruitment strategies to ensure the viability of progressing to a larger-scale study.

If the sample size is insufficient, the protocol will incorporate supplementary analyses, such as Bayesian methods, to provide more nuanced insights with limited data. Statistical tests for primary and secondary outcomes will be specified, with methods like logistic regression or survival analysis used as appropriate. To address the need for multiple comparisons, the protocol will apply correction techniques, such as Bonferroni adjustment or false discovery rate control, to ensure statistical rigour and minimise Type I errors.

### 6.10. Data Integrity

We will perform a thorough review of the data collection procedures to ensure adherence to the study protocol and verify the accuracy and completeness of the data. This will include an assessment of missing data, data consistency, and compliance with ethical and regulatory standards. If the data collection processes are found to be compliant with protocol, with minimal missing or inconsistent data, and overall data integrity is upheld, the study will proceed. If significant issues with data integrity are identified, such as widespread missing or inconsistent data, or non-compliance with ethical and regulatory standards, corrective actions will be taken. Persistent issues may lead to study termination if data integrity cannot be assured.

## 7. Preliminary Results

If the preliminary results show promising trends and outcomes consistent with the study hypotheses, and statistical analyses (including Bayesian methods if applicable) indicate the potential for meaningful findings, the study will continue. If the preliminary results are inconclusive or show no evidence of potential benefit, or if analyses indicate that the study objectives are unlikely to be met, the study will be reassessed. Decisions will be made based on the overall value of continuing versus terminating, considering the insights from the preliminary analyses.

To analyse of our outcomes of interest, logistic regression or linear mixed models, as well as survival analyses or chi-square tests, will be used.

### 7.1. Data Monitoring, Quality Control, and Assurance

The Chief Investigator (CI) will be responsible for the ongoing management of the study. The Sponsor will monitor and conduct audits on a selection of studies in their clinical research portfolio. Monitoring and auditing will be conducted in accordance with the UK Policy Framework for Health and Social Care 2017 and in accordance with the Sponsor’s monitoring and audit procedures.

### 7.2. Handling Participants Who Do Not Follow Their Allocated Arm

Participants who do not adhere to their allocated study arm will be tracked through the study’s monitoring system. Reasons for non-compliance will be documented, and alternative approaches will be considered to account for deviations in the final analysis. Non-compliance will also be recorded and assessed to understand its impact on study outcomes. The analysis will include intention-to-treat (ITT) and per-protocol approaches to evaluate the effect of treatment allocation versus the actual received treatment.

### 7.3. Recording and Assessing PGT-A Arm with No Embryos to Biopsy

For participants randomised to the PGT-A arm who have no embryos available for biopsy, detailed documentation will be maintained. This will include the reasons for the absence of viable embryos, such as poor ovarian response or embryo quality issues. These cases will be assessed and recorded as deviations from the intended intervention. The impact of such deviations on the study outcomes will be analysed, and the data will be included in the final study report to provide a comprehensive view of the feasibility and practical challenges of PGT-A’s implementation.

### 7.4. Ethical Considerations

All the necessary regulatory and ethical approvals were sought and received prior to the commencement of the study. All participating clinics are licenced by the Human Fertilisation and Embryology Authority (HFEA) in the UK, which permits PGT-A testing to be offered as described within the proposed research project. The study (Integrated Research Application System Number 236067) received approval from the Health Research Authority and Health and Care Research Wales (HCRW) and the East Midlands—Leicester South Research Ethics Committee (20/EM/0290). Any substantial amendments to the protocol will be submitted in writing by the CI to the REC, as well as the reasons for the amendment and the plan for future action.

### 7.5. Data Handling

Patient notes containing their personal and treatment details will be kept within the fertility units according to the statutory requirements of the HFEA Act 1990 [28] and the strict confidentiality that it requires, and only authorised staff at the study site will gain access to the case files. Notes from patients who achieved a pregnancy will be kept or archived for 50 years. Patients will need to provide written consent before any of their treatment details or personal information is passed on to their General Practitioners or any other persons who are not covered by an HFEA licence.

### 7.6. Data Management and Oversight

Site investigators will take responsibility for the conduct of the study and site investigators will supervise the day-to-day operation of the project and are responsible for ensuring that Good Clinical Practice guidelines are followed. Members of the research team from King’s College London will monitor the data.

### 7.7. Report and Dissemination

The trial will be registered at http://www.clinicaltrials.gov and will also be registered on the International Standard Randomised Controlled Trial Number (ISRCTN) Register at https://www.isrctn.com. The results of the research will be reported via an internal report and submitted to the regulatory authorities. We also aim to disseminate the results through conference presentations and peer-reviewed journals.

## 8. Results

In our study protocol, the results will be presented with a focus on clarity and precision. Descriptive statistics, including percentages and means with standard deviations, will be used to summarise the data. To provide additional insight into the precision and uncertainty of our findings, we will also report confidence intervals (CIs) for key outcomes. CIs will help elucidate the clinical significance of our results, offering a range within which the true effect is likely to fall. This comprehensive approach will ensure that the findings are both statistically rigorous and clinically meaningful.

## 9. Discussion

IVF is the leading treatment for infertility. However, the success rate remains largely unchanged despite ongoing laboratory and ovarian stimulation advances, with worsening prognoses seen to be associated with advances in the patient’s age [28]. PGT-A using NGS is currently being suggested as a technique to improve embryo selection prior to transfer, and therefore pregnancy outcomes, in women undergoing IVF. While past studies have previously investigated the benefit of PGT-A, these studies have been criticised as they did not assess the clinical efficacy of NGS, which is what most assisted-conception clinics have adopted, which allows for an analysis of all 24 chromosomes [14].

Worldwide, an increasing number of live births are being reported following the transfer of mosaic embryos; however, previous studies did not describe any follow-up with a foetal medicine unit [29,30]. Through our study, we aim for all eligible embryos to undergo biopsy and to subsequently be ranked based on their individual implantation potential and appropriately recommended for transfer—namely, euploid embryos followed by low-mosaic, high-mosaic and aneuploid embryos. No studies have assessed the clinical pregnancy and live birth rates following the transfer of low-mosaic embryos. In our study, an obstetrics follow-up within our foetal medicine unit for mosaic embryos will be offered.

With the increase in the aneuploidy rates expected in our age group of interest, the pilot study and proposed RCT is designed to address the gaps in the utility of PGT-A as a treatment for subfertility in patients aged between 35 and 42 years. If the beneficial effect of this intervention can be confirmed in this age group, PGT-A will provide a cost-effective method for helping couples to conceive. On the other hand, if PGT-A is not found to have a superior effect or is proved to be detrimental to the probability of conception, the use of this procedure can be abandoned by clinics and doctors who are recommending this procedure to their patients.

### Strengths and Weaknesses of This Study

This pilot study and the proposed RCT will provide the first comprehensive study to assess whether IVF with PGT-A is more clinically effective, safer, and more cost-effective than IVF without PGT-A when providing IVF in women of advanced reproductive age seeking IVF.Our study design will allow us to rank embryos based on implantation potential, including the transfer of mosaic embryos, defining the clinical pregnancy rates associated with these.The primary outcomes of this study will be to assess recruitment and adherence to the study protocol, and secondary outcomes will include the clinical pregnancy rate, miscarriage rate, and an analysis of time to live birth.The lack of blinding of the clinicians and patients is a limitation to the study design.

## Figures and Tables

**Figure 1 jcm-13-06192-f001:**
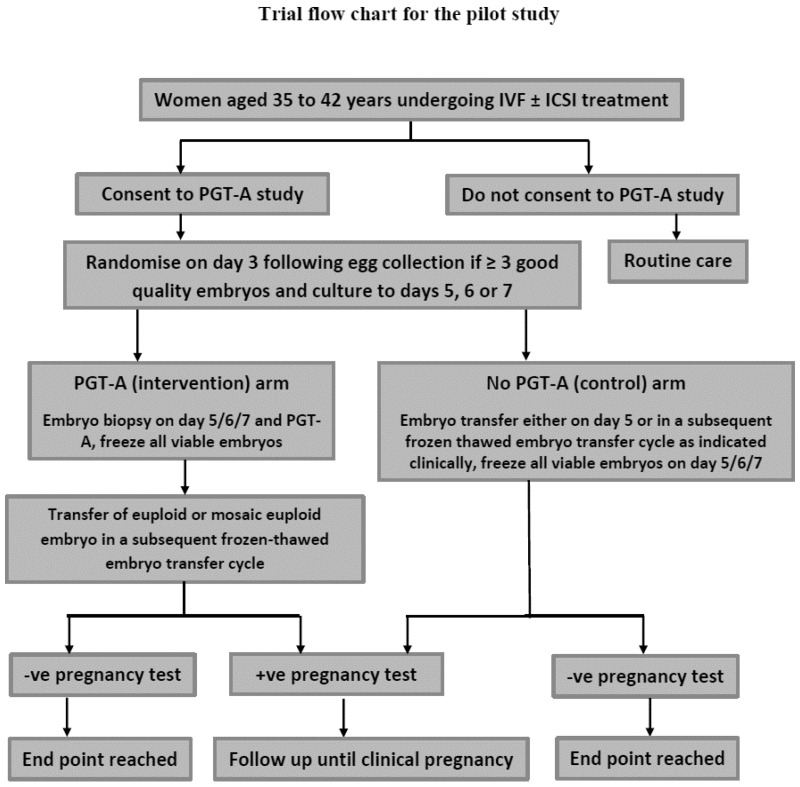
Participant flow chart through PGT-A study.

## Data Availability

No new data were created or analyzed in this study.

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
