# Peer review of "Preimplantation Genetic Testing for Aneuploidy (PGT-A) in In-Vitro Fertilisation (IVF) Treatment: Study Protocol for Pilot Phase of a Randomised Controlled Trial"

_jcm, 2024, doi:10.3390/jcm13206192_

Round 1
Reviewer 1 Report
Comments and Suggestions for Authors
Thank you for submitting your research document on "Preimplantation Genetic Testing for Aneuploidy (PGT-A) in In Vitro Fertilization (IVF) Treatment." Overall, the document is logically clear, the study design is reasonable, and it has certain innovations and clinical significance. However, there are still some areas that need further improvement and clarification to enhance the quality and readability of the research. Here are my specific comments:
Sample Size Calculation:
The document should provide the specific formula and assumptions used for sample size calculation to help reviewers and readers understand the rationale behind the sample size.
Recruitment Criteria:
Setting a standard to recruit at least 60 women is reasonable, but further explanation is needed as to why this specific number was chosen. Is there literature supporting the validity of this sample size? If the sample size is insufficient, how will this affect the interpretation and generalizability of the results?
Analysis Plan:
The document should clearly outline how data and results will be handled in the case of insufficient sample size. Is there a plan for supplementary data analysis or the use of alternative methods (e.g., Bayesian analysis) to address the issue of insufficient sample size? Additionally, the document should specify which statistical tests will be used to analyze the primary and secondary outcomes. If multiple statistical tests are conducted, it should explain how to control for multiple comparison issues.
Presentation of Results:
In addition to percentages and means (standard deviation), consideration should be given to using confidence intervals (CI) to provide precision and uncertainty of the results. This will help readers better understand the clinical significance of the findings.
Interim Analysis:
The document should specify the content of the interim analysis, including the assessment of recruitment progress, data integrity, and preliminary results.
Stopping Criteria:
Clear criteria need to be established to determine whether to continue or terminate the study.
Randomization Method:
The document mentions using "third-party, remote, internet-based block randomization with minimization." Detailed implementation steps of this method are needed, including how to ensure the confidentiality of randomization and the concealment of allocation. For example, the specific algorithms used for randomization, the software or tools employed, and how to prevent researchers from influencing participant selection during allocation.
Blinding Design:
Although the document states that blinding allocation is not feasible due to the nature of the PGT-A strategy and relevant regulations, it is still necessary to discuss how to minimize bias from researchers and participants. For instance, will other measures (such as blinding data analysts) be taken to reduce potential bias?
Recruitment and Informed Consent:
The recruitment process for participants should be clearly outlined, ensuring that all participants fully understand the purpose, process, and potential risks of the study before participation. The process of obtaining informed consent should comply with ethical standards and be described in detail in the document.
Author Response
1. Summary
Thank you very much for taking the time to review this manuscript. Please find the detailed responses below and the corresponding revisions/corrections highlighted/in track changes in the re-submitted files
Thank you for submitting your research document on "Preimplantation Genetic Testing for Aneuploidy (PGT-A) in In Vitro Fertilization (IVF) Treatment." Overall, the document is logically clear, the study design is reasonable, and it has certain innovations and clinical significance. However, there are still some areas that need further improvement and clarification to enhance the quality and readability of the research. Here are my specific comments:
Sample Size Calculation:
The document should provide the specific formula and assumptions used for sample size calculation to help reviewers and readers understand the rationale behind the sample size.
Thank you for your comment.
For this pilot feasibility randomised controlled trial, a sample size of 100 women has been determined. This sample size was chosen as it allows us to estimate recruitment rates with a margin of error of ±10.5% around the true rate, providing an 80% confidence interval. While formal hypothesis testing is not the primary focus of this pilot study, an alpha value of 0.05 (5%) and a beta value corresponding to 80% power (β = 0.2) would be considered standard if preliminary hypothesis testing were conducted. Given the strict regulatory framework governing IVF in the UK, particularly the mandatory reporting of all IVF treatment outcomes as required by the Human Fertilisation and Embryology Authority (HFEA), we anticipate nearly complete follow-up, approaching 100%.
This has been added in the manuscript (Page 14, Lines 352-361)
Recruitment Criteria:
Setting a standard to recruit at least 60 women is reasonable, but further explanation is needed as to why this specific number was chosen. Is there literature supporting the validity of this sample size? If the sample size is insufficient, how will this affect the interpretation and generalizability of the results?
Thank you for your comment and for seeking clarification on the recruitment benchmarks. The threshold of 60 women out of the planned 100 participants has been selected as a critical benchmark for evaluating the feasibility of progressing to the main study. This represents 60% of the total target sample size, which we consider a sufficient indicator of recruitment success within the given 18-month timeframe.
Reaching this target would demonstrate an adequate recruitment rate, suggesting that the study is likely to meet its enrolment goals within a reasonable period if expanded to a larger scale. Conversely, failing to recruit at least 60 participants would signal potential challenges in participant recruitment, necessitating a re-assessment of recruitment strategies or study design before proceeding to the main study. This approach ensures that resources are used efficiently and that the main study is only launched if there is a reasonable expectation of achieving full recruitment.
This has been added in the manuscript (Page 14, Lines 373-383)
Analysis Plan:
The document should clearly outline how data and results will be handled in the case of insufficient sample size. Is there a plan for supplementary data analysis or the use of alternative methods (e.g., Bayesian analysis) to address the issue of insufficient sample size? Additionally, the document should specify which statistical
tests will be used to analyse the primary and secondary outcomes. If multiple statistical tests are conducted, it should explain how to control for multiple comparison issues.
Thank you for your valuable feedback. We appreciate the suggestion to outline how data and results will be managed in the case of insufficient sample size. To address this, we have incorporated how supplementary analyses, including Bayesian methods can be used to handle limited data effectively. In the event of an insufficient sample size, the study protocol now includes clear plan for handling data and results. This plan includes the potential use of supplementary data analysis methods, such as Bayesian analysis, which can provide more flexible and informative insights even with limited data.
For analysing the primary and secondary outcomes, specific statistical tests will be identified in the protocol. For instance, primary outcomes may be analysed using logistic regression or linear mixed models, depending on the nature of the data, while secondary outcomes may involve survival analysis or chi-square tests, as appropriate.
If multiple statistical tests are conducted, the choice of method will depend on the number of comparisons and the nature of the data, ensuring that the study’s conclusions remain statistically robust.
This has been added in the manuscript (Page 15, Lines 393-400)
Presentation of Results:
In addition to percentages and means (standard deviation), consideration should be given to using confidence intervals (CI) to provide precision and uncertainty of the results. This will help readers better understand the clinical significance of the findings.
Thank you for the suggestion. We will be incorporating confidence intervals (CIs) alongside percentages and means (with standard deviations) to allow the readers to better gauge the clinical significance of the findings. We will ensure that CIs are included in our statistical analyses to improve the interpretability and robustness of the reported outcomes.
This has been added in the manuscript (Page 17, Lines 446-451)
Interim Analysis:
The document should specify the content of the interim analysis, including the assessment of recruitment progress, data integrity, and preliminary results.
Thank you for your helpful comment. We have now included the following in the manuscript:
1.Recruitment Progress: We have explained that we will evaluate whether at least 60 out of 100 participants have been recruited within the 18-month period. This threshold was a consensus reached by the team as test of feasibility for proceeding to the main study. If recruitment is below this target, we will reassess and adjust our recruitment strategies as needed.
2.Data Integrity: We have explained how we will review data collection processes to ensure adherence to protocol and verify that data is accurate and complete. This will include checking for missing or inconsistent data and confirming compliance with regulatory and ethical standards.
3.Preliminary Results: We have now also explained how we will conduct preliminary analyses to assess early trends and outcomes. This may involve descriptive statistics and, if necessary, supplementary Bayesian analyses to interpret the limited data effectively.
Thank you for suggesting these addition to our manuscript.
This has been added in the manuscript (Page 14-15, Lines 372-400)
Stopping Criteria:
Clear criteria need to be established to determine whether to continue or terminate the study.
Thank you for highlighting the importance of establishing clear criteria for determining whether to continue or terminate the study.
We have now incorporated explicit criteria into our interim analysis plan in our manuscript as described above. In summary, we will assess recruitment progress by determining whether at least 60 out of 100 participants have been enrolled within the 18-month period, using this benchmark to evaluate the feasibility of moving forward with the main study. If recruitment falls short of this target, we will reassess and modify
our recruitment strategies accordingly.
This has been added in the manuscript (Page 14-15, Lines 372-383)
Randomisation Method:
The document mentions using "third-party, remote, internet-based block randomisation with minimization." Detailed implementation steps of this method are needed, including how to ensure the confidentiality of randomisation and the concealment of allocation. For example, the specific algorithms used for randomisation, the software or tools employed, and how to prevent researchers from influencing participant selection during allocation.
Thank you for your comment.
We have included a revised version of the implementation steps for the randomisation process, incorporating details on confidentiality, allocation concealment, and specific algorithms:
Randomisation Process:
To ensure rigorous randomisation and complete allocation concealment, we will use third-party, remote, internet-based block randomisation with minimization. The implementation will include the following detailed steps:
1.Randomisation System: The randomisation will be managed by an independent, third-party system that ensures a 1:1 allocation ratio. The system will use a minimization algorithm to balance key factors among study groups.
2.Algorithm and Software: Minimisation Algorithm: The algorithm will be designed to minimize imbalances across several factors, including IVF clinic, female age (35-37, 38-39, 40-42 years), BMI (20 - <25, 25 - <30, ≥30), previous live birth, type of ovarian stimulation protocol (GnRH agonist or antagonist, hCG or GnRH agonist trigger), and standard IVF or ICSI.
Software/Tools: We will use MedSciNet Clinical Trial Framework (www.medscinet.net), a secure and validated tool for clinical trials. This software ensures confidentiality of randomisation and prevents researchers from influencing participant selection.
3.Confidentiality and Allocation Concealment: Access Control: Only authorized personnel will have access to the randomisation system. The system will be designed to prevent unauthorized access and manipulation.
Concealment: The randomisation and allocation process will be fully automated to prevent any researcher involvement in participant assignment. The allocation sequence will be concealed and only revealed to the clinical team at the time of participant assignment.
By adhering to these detailed procedures, we will maintain the integrity of randomisation and ensure unbiased participant allocation throughout the study.
This has been added in the manuscript (Page 9, Lines 237-255)
Blinding Design:
Although the document states that blinding allocation is not feasible due to the nature of the PGT-A strategy and relevant regulations, it is still necessary to discuss how to minimize bias from researchers and participants. For instance, will other measures (such as blinding data analysts) be taken to reduce potential bias?
Thank you for your comment. Although blinding allocation is not feasible due to the nature of the PGT-A strategy and relevant regulations, we will implement several measures to minimize potential bias from both researchers and participants:
1.Blinding of Data Analysts: Data analysts will be blinded to participant allocation to prevent any bias in the interpretation of results. This will ensure that the analysis is conducted impartially, regardless of the group to which participants have been assigned.
2.Standardisation of Procedures: All study procedures, including data collection and outcome assessment, will be standardised to minimise variability and reduce the potential for bias. Protocols will be strictly followed, and training will be provided to ensure consistency across all study sites.
3.Independent Monitoring: Regular audits and reviews will be conducted to maintain the integrity of the study.
4.Blinded Outcome Assessment: Wherever possible, outcome assessors will be blinded to participant allocation to prevent their knowledge of the allocation from influencing their assessment of outcomes.
5.Confidentiality and Data Security: We will implement robust measures to secure participant data and maintain confidentiality, further reducing the risk of bias. Access to sensitive information will be restricted to authorized personnel only.
We have included these strategies in the updated protocol to address potential concerns and uphold the study's integrity.
This has been added in the manuscript (Page 10, Lines 262-276)
Recruitment and Informed Consent:
The recruitment process for participants should be clearly outlined, ensuring that all participants fully understand the purpose, process, and potential risks of the study before participation. The process of obtaining informed consent should comply with ethical standards and be described in detail in the document.
Thank you for your comment. We have revised the section on the recruitment process and informed consent, incorporating detailed procedures to ensure clarity and compliance with ethical standards: Below is a summary of our proposed recruitment process and this is now reflected in the manuscript.
Recruitment Process and Informed Consent
Recruitment Process:
Participants will be recruited during their initial consultation at the fertility clinic. Recruitment will continue through monitoring scans up to the day of egg collection. Detailed steps in the recruitment process include:
1.Initial Consultation: During the initial consultation, patients will be provided with comprehensive information about the study. This includes the purpose of the study, the study procedures, and potential risks and benefits.
2.Information Provision: Patients will receive a detailed study information leaflet outlining the study objectives, methods, potential risks, and benefits. This information will be designed to ensure that participants fully understand the study before agreeing to participate.
3.Ongoing Recruitment: Patients will be continuously informed about their participation status during monitoring scans and other clinical interactions until the day of egg collection. This ensures that participants remain fully informed throughout the recruitment process.
4.Additional Centres: If the study progresses to its full implementation, additional recruitment centres may be involved. These centres will follow the same recruitment and consent procedures to maintain consistency and adherence to ethical standards.
Informed Consent:
1.Consent Process: The process of obtaining informed consent will be conducted in accordance with ethical standards. Patients will be given sufficient time to review the study information and ask any questions before making a decision.
2.Consent Form: Patients will be required to sign an informed consent form, which will document their understanding of the study and their willingness to participate. This form will include details on the study purpose, procedures, potential risks, benefits, and their right to withdraw at any time without affecting their standard care.
3.Documentation: All consent forms will be securely stored and managed to ensure confidentiality and compliance with data protection regulations. Records of consent will be maintained and available for review by relevant oversight bodies.
The above has been reflected throughout the manuscript.

Reviewer 2 Report
Comments and Suggestions for Authors
This is a publication of a study protocol for a pilot randomized control trial assessing IVF with PGT-A with NGS to IVF without PGT-A in infertility patients age 35-42. This study is a welcome addition to the limited literature assessing the utility of PGT-A,
Under the "Strengths and Weaknesses of the study" in lines 35-44, it is recommended that the authors state what they are comparing PGT-A to (eg IVF without PGT-A). Furthermore, the statement "Our study design will allow us to rank embryos based on implantation potential including the transfer of mosaic embryos, defining the clinical pregnancy rates associated with these." seems overstated given this is a pilot study.
The introduction is unnecessarily long, particularly the paragraphs on morphologic assessment in IVF and could be easily shortened to focus the introduction on the previous PGT-A studies. Similarly, there is little review of the outcomes of the previous PGT-A studies which did not show benefit and a clearer description of how this study differs from those previously published reports is needed in the introduction.
Grammatical errors are noted throughout including missing commas. Please do a thorough grammatical review.
The randomisation with minimisation algorithm appears to have so many factors, that I worry this is not feasible. Can you please specifically describe how this will be done? Furthermore, when will randomisation occur? How will you measure patients who do not end up following their allocated study arm? How will you record and assess patients who were randomised to the PGT arm who end up having no embryos to biopsy?
What is the definition of low mosaic and high mosaic embryos (eg. what cut offs will be used)?
"Mixing of low mosaic with euploid embryos in one transfer event is not recommended." Does this mean it is allowable? Similarly, if a patient chooses to have a double embryo transfer in either study arm, will they be excluded from the analysis or how will this be accounted for?
If patients choose not to transfer mosaic embryos, how will be these recorded and incorporated into results?
It is unclear if the end point of the pilot study is a negative pregnancy test after the transfer of the first embryo or only after the transfer of all embryos from that cycle - please clarify how long patients will be followed (ie. until all embryos have been transfered or until a certain time point for example).
This study has merit, but given this is a publication of the study protocol, more elaboration is needed on the details of the study protocol.
Comments on the Quality of English LanguageSome run on sentences, missing commas, missing conjunctions and missing comparisons throughout the manuscript.
Author Response
For research article
|
Response to Reviewer 2 Comments
|
||
|
1. Summary |
|
|
|
Thank you very much for taking the time to review this manuscript. Please find the detailed responses below and the corresponding revisions/corrections highlighted/in track changes in the re-submitted files. |
||
|
This is a publication of a study protocol for a pilot randomized control trial assessing IVF with PGT-A with NGS to IVF without PGT-A in infertility patients aged 35-42. This study is a welcome addition to the limited literature assessing the utility of PGT-A, |
|
Under the "Strengths and Weaknesses of the study" in lines 35-44, it is recommended that the authors state what they are comparing PGT-A to (eg IVF without PGT-A). |
|
Thank you for this comment. This has been appropriately clarified. |
|
Furthermore, the statement "Our study design will allow us to rank embryos based on implantation potential including the transfer of mosaic embryos, defining the clinical pregnancy rates associated with these." seems overstated given this is a pilot study. |
|
Thank you for your comment. You recognise the potential overstatement in the statement regarding embryo ranking and clinical pregnancy rates. As a pilot study, our primary aim is to assess feasibility and refine methods, rather than to establish definitive clinical outcomes.
The following revised statement has been included in the manuscript. In the context of this pilot study, our design will facilitate the evaluation of embryo ranking based on implantation potential, including the assessment of mosaic embryos. However, given the exploratory nature of the pilot, our findings will be preliminary and focused on feasibility. We aim to gather initial data that will inform future studies, rather than drawing definitive conclusions about clinical pregnancy rates at this stage.
We hope that you agree that this revision clarifies that the pilot study is intended to provide preliminary insights and set the stage for more comprehensive research in subsequent phases.
This has been included in our manuscript (Page 7, Line 94-98)
|
|
The introduction is unnecessarily long, particularly the paragraphs on morphologic assessment in IVF and could be easily shortened to focus the introduction on the previous PGT-A studies. Similarly, there is little review of the outcomes of the previous PGT-A studies which did not show benefit and a clearer description of how this study differs from those previously published reports is needed in the introduction. |
|
Thank you for your feedback.
We have revised our approach for the introduction and reduced the word count significantly. We have also focused on the relevant aspects of previous PGT-A studies and clearly delineates how our study differs from past research. In particular, we discussed how in the STAR study, using trophectoderm biopsy and NGS-based PGT-A for selecting euploid embryos, there was significantly increase in ongoing pregnancy rates (OPR) per cycle compared to non-PGT-A methods. We also discussed how this study involved women with an average age of 28 and while a post hoc analysis of women aged 35–40 years indicated a notable increase in OPR per embryo transfer with PGT-A, this finding was debated and not considered reliable by the HFEA's SCAAC. We also expanded on other studies, including Yan et al. (2021), which involved 1,212 patients and an average age of 29, which also failed to show a significant advantage of PGT-A over non-PGT-A methods in cumulative live-birth rates. We also discussed how the study faced criticisms for methodological flaws, such as selecting only three top-quality embryos for biopsy and not accounting for the transfer of low-grade mosaic embryos. We also discussed how ESHRE now recognises that cumulative live birth rates may not accurately reflect PGT-A’s superiority due to biases and differences in embryo selection and transfer practices.
|
|
Grammatical errors are noted throughout including missing commas. Please do a thorough grammatical review. |
|
Thank you for pointing that out. We appreciate your attention to details and all grammatical errors, including missing commas, have been thoroughly corrected. |
|
The randomisation with minimisation algorithm appears to have so many factors, that I worry this is not feasible. Can you please specifically describe how this will be done? Furthermore, when will randomisation occur? How will you measure patients who do not end up following their allocated study arm? How will you record and assess patients who were randomised to the PGT arm who end up having no embryos to biopsy? |
|
Thank you for your comment.
Randomisation System:
The above changes are included in our ammended manuscript (Page 9, Line 238-247)
Factors for Balancing: The minimization algorithm will consider the following critical factors:
The above changes are included in our ammended manuscript (Page 10, Line 251-255)
Algorithm Functionality:
System Operation:
Timing of Randomisation: Randomisation will occur on day 3 following egg collection. At this point, all consented patients will be evaluated for eligibility. Specifically, patients will be randomised if they have at least three good-quality embryos, as assessed using the embryo scoring system of the Association of Clinical Embryologists (UK).
The above changes are included in our ammended manuscript (Page 11, Line 292-294)
Care Pathway Post-Randomisation:
The above changes are included in our ammended manuscript (Page 11, Line 296-299)
Handling Participants Who Do Not Follow Their Allocated Arm:
The above changes are included in our ammended manuscript (Page 15, Line 402-407)
Recording and Assessing PGT-A Arm with No Embryos to Biopsy:
The above changes are included in our ammended manuscript (Page 15, Line 408-414)
By following these procedures, we aim to ensure that the randomisation process is both feasible and robust, while also adequately addressing and documenting any issues that arise during the study.
The above changes are included in our ammended manuscript. |
|
What is the definition of low mosaic and high mosaic embryos (e.g. what cut offs will be used)? |
|
Thank you for pointing out this lack of definition. The manuscript has now been updated. For our study, a low mosaic Embryos was defined as an embryo with whole chromosome aneuploidies in 30-50% of the biopsied cell and high mosaic embryo was defined as an embryo with whole chromosome aneuploidies in 50-70% of the biopsied cells For chromosome numbers 13, 18, 21 and the sex chromosomes, embryos are considered to be suitable for transfer if 70% or more of the biopsied cells have the correct number of chromosomes.
The above changes are included in our ammended manuscript (Page 11, Line 303-305)
|
|
"Mixing of low mosaic with euploid embryos in one transfer event is not recommended." Does this mean it is allowable? Similarly, if a patient chooses to have a double embryo transfer in either study arm, will they be excluded from the analysis or how will this be accounted for? |
|
Thank you for your comment. To clarify, mixing low mosaic embryos with euploid embryos in a single transfer event is not recommended and the current best practice and recommendation for this study is to perform single embryo transfers to optimise the chances of a successful outcome and to maintain the clarity of the study results. The manuscript has been updated to re-iterate this.
The above changes are included in our ammended manuscript (Page 12, Line 307-311) |
|
If patients choose not to transfer mosaic embryos, how will be these recorded and incorporated into results? |
|
Thank you for your comment and for raising the need for this clarification.
If patients choose not to transfer mosaic embryos, these decisions will be carefully recorded and accounted for in the study analysis. Specifically:
By incorporating these factors into the analysis, the study aims to provide a comprehensive view of the implications of transferring versus not transferring mosaic embryos and to ensure that the findings are robust and applicable to various clinical scenarios.
The above changes are included in our ammended manuscript (Page 12,Line 315-321)
|
|
It is unclear if the end point of the pilot study is a negative pregnancy test after the transfer of the first embryo or only after the transfer of all embryos from that cycle - please clarify how long patients will be followed (i.e.. until all embryos have been transferred or until a certain time point for example). |
|
Many thanks for raising this confusion.
The above changes are included in our ammended manuscript (Page 12-13, Line 322-326) |
|
This study has merit but given this is a publication of the study protocol, more elaboration is needed on the details of the study protocol. |
|
Thank you for your feedback. We appreciate your recognition of the study's merit. As this is a publication of the study protocol, we acknowledge the need for more detailed elaboration. We have enhanced the protocol to provide comprehensive information on:
We have incorporated these details to provide a clearer and more comprehensive view of the study protocol.
|

Reviewer 3 Report
Comments and Suggestions for Authors
The authors proposed a study protocol on the effectiveness of the PGT-A on the clinical outcomes in IVF. The topic is well described in literature and, as presented in its current form, lacks in novelty.
- The authors should shorten the introduction and rewrite it more focused on the topic. It is not clear the aim of the study and what would add to the current knowledge
- IRB is mandatory for a RCT, please provide evidence
- Please provide a power analysis to define the numer of patients to enroll
- The authors should clarify the rationale of including low mosaic embryos and the number of them considered in the analysis
- Extensive english revision is recommended together with punctuations correnctions
Comments on the Quality of English LanguageExtensive editing of English language required.
Author Response
1. Summary
Thank you very much for taking the time to review this manuscript. Please find the detailed responses below and the corresponding revisions/corrections highlighted/in track changes in the re-submitted files
The authors proposed a study protocol on the effectiveness of the PGT-A on the clinical outcomes in IVF. The topic is well described in literature and as presented in its current form, lacks in novelty.
Thank you for your feedback.
We appreciate your insights regarding the study's novelty.
We recognise that while the topic of PGT-A in IVF is well-established in the literature, our study aims to address specific gaps and provide new insights. To enhance the protocol and demonstrate its contribution, we have updated our proposed manuscript to focus on the following aspects:
1.
Novelty and Unique Aspects: Highlighting how our study differs from previous research, including any innovative methodologies, specific population groups, or new outcomes being assessed.
Our study aims to address significant gaps in the current literature on PGT-A by focusing on a higher age group and evaluating new methods with a particular emphasis on mosaic embryos. While past research, including four RCTs (Schoolcraft et al., 2012; Yang et al., 2012; Forman et al., 2013; Scott et al., 2013), has explored new PGT-A methods, these studies were limited by small sample sizes and methodological issues, including insufficient power to detect differences in live birth rates—a crucial primary outcome.
Recent meta-analyses have reported improvements in implantation rates (IR) with new PGT-A methods (Dahdouh et al., 2015; Chen et al., 2015), but these analyses were criticized for focusing on IR rather than intention-to-treat (ITT) outcomes. The multi-centre RCT by Munne et al. (2017) found no overall difference in ongoing pregnancy rates between PGT-A and standard morphology, though a subgroup analysis indicated benefits for women aged 35–40 years.
Our study's novelty lies in its focus on an older age group, where the prevalence of aneuploidy and mosaicism is higher, and evaluating the impact of advanced PGT-A methods on live birth rates. Additionally, we will specifically address the outcome of mosaic embryos—a critical aspect often overlooked in previous studies. By examining how mosaic embryos affect clinical outcomes and incorporating them into the analysis, our study will provide valuable insights into the practical implications of PGT-A for diverse embryo classifications.
By addressing these aspects, we aim to offer new evidence on the effectiveness of PGT-A in improving IVF outcomes, particularly for older women and in the context of mosaic embryo management.
This has been included in our manuscript (Page 6-7, Line 148-180)
2.
Specific Research Questions: Clearly articulating the unique research questions or hypotheses that our study addresses and how these may contribute to advancing knowledge in the field.
Our study is designed to address several key research questions and hypotheses that are critical for advancing the understanding and application of PGT-A in IVF, particularly for women of advanced reproductive again and in the context of mosaic embryos:
•
Effectiveness of PGT-A in women of advanced reproductive age
o
Research Question: How does PGT-A impact live birth rates in women aged 35-42 compared to standard IVF procedures?
o
Hypothesis: PGT-A will demonstrate a significant improvement in live birth rates among women aged 35-42, compared to standard IVF protocols. This hypothesis is based on the premise that women of advanced reproductive age have a higher incidence of chromosomal abnormalities, and that PGT-A can help select embryos with a higher likelihood of successful implantation and live birth.
•
Impact of Mosaic Embryos on Outcomes:
o
Research Question: What is the effect of including mosaic embryos in embryo transfer cycles on clinical outcomes, such as implantation rates, ongoing pregnancy rates, and live birth rates?
o
Hypothesis: The inclusion of mosaic embryos in the transfer process will not adversely affect the overall clinical outcomes compared to the exclusion of these embryos. We hypothesise that careful management and selection of mosaic embryos may yield similar or improved outcomes compared to transfers with only euploid embryos.
This has been included in our manuscript (Page 8, Line 200-207)
3.
Detailed Protocol Enhancements: Expanding on any novel aspects of the study design, such as advanced techniques or specific interventions, to underline the study's potential impact.
Our study incorporates several novel aspects and advanced techniques that set it apart from previous research and have the potential to significantly impact the field of reproductive medicine. Here, we outline these key enhancements in detail:
1.
Advanced PGT-A Techniques:
o
Next-Generation Sequencing (NGS): The study will utilise advanced NGS-based PGT-A methods, which offer superior resolution and accuracy in identifying chromosomal abnormalities compared to traditional methods. NGS enables comprehensive analysis of all 24 chromosomes, improving the detection of aneuploidy and mosaicism.
o
Mosaicism Detection: Our protocol includes a focused investigation into the clinical implications of mosaic embryos. Unlike earlier studies, which often excluded mosaic embryos from analysis, our study will assess how different levels of mosaicism impact embryo viability and overall IVF outcomes.
2.
Inclusion of Older Age Groups:
o
Focus on Advanced Reproductive Age: The study targets women aged 35-42, a group often underrepresented in previous research. This demographic is particularly relevant due to its higher incidence of chromosomal abnormalities. By focusing on this age group, we aim to provide targeted insights into how PGT-A can benefit older women, where traditional IVF approaches may fall short.
3.
Randomisation and Allocation Strategy:
o
Comprehensive Randomisation Protocol: We will use a sophisticated randomisation process involving a minimisation algorithm to ensure balanced allocation across key factors, such as IVF clinic, female age, BMI, previous live birth, type of ovarian stimulation protocol, and type of IVF procedure. This approach minimizes potential biases and enhances the robustness of the study results.
o
Real-Time Data Management: The randomisation system will be monitored and adjusted in real-time to ensure accurate and unbiased assignment of participants to study arms.
4.
Single Embryo Transfer Protocol:
o
Exclusive Single Embryo Transfers: The study will enforce a protocol of single embryo transfer to optimize outcomes and reduce risks associated with multiple pregnancies. This approach aligns with current best practices and allows for a more precise assessment of PGT-A’s efficacy in a controlled setting.
5.
Patient-Centric Approach and Consent:
o
Detailed Informed Consent Process: We will implement a thorough informed consent process, ensuring that participants fully understand the study's objectives, procedures, and potential risks. This includes providing clear information about the implications of embryo mosaicism and the rationale behind study interventions.
o
Patient Preferences Documentation: The study will document and analyse patient preferences regarding the transfer of mosaic embryos. This information will help evaluate how patient choices influence clinical decisions and outcomes, adding a layer of patient-centred research to the study.
6.
Outcome Measurement and Analysis:
o
Enhanced Outcome Metrics: In addition to traditional metrics such as implantation rates and live birth rates, we will include comprehensive measures of embryo quality and mosaicism levels. This detailed approach will provide a nuanced understanding of how advanced PGT-A techniques affect clinical outcomes.
o
Intention-to-Treat Analysis: To address potential biases, the study will utilize intention-to-treat (ITT) analysis alongside per-protocol analysis. This dual approach will ensure that all randomized participants are included in the analysis, regardless of whether they adhered to the allocated study arm.
By integrating these detailed protocol enhancements, our study aims to address critical gaps in the current literature, provide new insights into PGT-A’s effectiveness, and contribute to the optimisation of IVF practices. These advancements will help refine clinical guidelines and improve patient outcomes in reproductive medicine.
This has been included in our manuscript
- The authors should shorten the introduction and rewrite it more focused on the topic. It is not clear the aim of the study and what would add to the current knowledge
Thank you for your feedback.
We have revised our approach for the introduction and reduced the word count significantly. We have also focused on the relevant aspects of previous PGT-A studies and clearly delineates how our study differs from past research.
- IRB is mandatory for a RCT, please provide evidence
Thank you for your comment. The study was successful through its IRAS application (Project ID: 236067), its Research and Ethics committee review (REC reference: 20/EM/0290). It was also approved by the NHS Health and Research Authority (HRA and Health and Care Research Wales).
This has been added to the manuscript.
- Please provide a power analysis to define the number of patients to enrol
Thank you for this comment.
For this pilot feasibility randomised controlled trial, a sample size of 100 women has been determined. This sample size allows us to estimate recruitment rates with a margin of error of ±10.5% around the true rate, providing an 80% confidence interval. While formal hypothesis testing is not the primary focus of this pilot study, an alpha value of 0.05 (5%) and a beta value corresponding to 80% power (β = 0.2) would be considered standard if preliminary hypothesis testing were conducted. Given the strict regulatory framework governing IVF in the UK, particularly the mandatory reporting of all IVF treatment outcomes as required by the Human Fertilisation and Embryology Authority (HFEA), we anticipate nearly complete follow-up, approaching 100%.
This has been included in our manuscript (Page 13, Line 347-356)
- The authors should clarify the rationale of including low mosaic embryos and the number of them considered in the analysis
Thank you for your comment and for raising this important point.
The inclusion of low mosaic embryos in the study is driven by the need to understand their potential impact on IVF outcomes more comprehensively. Historically, many IVF studies have excluded mosaic embryos due to concerns about their viability and potential for lower success rates. However, recent advancements in embryo screening and management suggest that low mosaic embryos may still offer valuable clinical benefits.
Given the exploratory nature of the pilot, our findings will be preliminary and focused on feasibility. However, we aim to gather initial data that will inform our future RCT study, rather than drawing definitive
conclusions about clinical pregnancy rates at this stage.
Our planned RCT will focus on:
1.
Expanding Knowledge on Embryo Viability: Investigating whether low mosaic embryos can positively contribute to pregnancy outcomes compared to euploid embryos.
2.
Improving Clinical Practice: Providing data on how low mosaic embryos affect IVF success rates and refining guidelines for embryo selection and transfer.
3.
Addressing Patient Preferences: Reflecting real-world practices where mosaic embryos are sometimes included in transfers.
In our study, a low mosaic embryo is defined as an embryo with whole chromosome aneuploidies in 30-50% of the biopsied cell and a high mosaic embryo is defined as an embryo with whole chromosome aneuploidies in 50-70% of the biopsied cells.
The above has been included in our manuscript (Page 12, Line 303-305).
The number of low mosaic embryos considered will depend on their availability and mosaicism level per patient, with detailed documentation of their numbers and outcomes. The analysis will evaluate implantation rates, ongoing pregnancy rates, and live birth rates for cycles involving low mosaic embryos, compared to those with euploid and higher mosaicism embryos. Advanced statistical methods will be employed to assess the impact of low mosaic embryos on clinical outcomes, including subgroup analyses to compare their effects with euploid embryos.
By clarifying the rationale and methodology for including low mosaic embryos, the study aims to address critical gaps in current knowledge and provide valuable insights into their role in IVF success. This approach will contribute to a more comprehensive understanding of embryo selection and its implications for reproductive outcomes.
The above has been included in our manuscript.
- Extensive English revision is recommended together with punctuations corrections
Thank you for pointing that out.
We appreciate your attention to details and all grammatical errors, including missing commas, have been thoroughly corrected.

Round 2
Reviewer 1 Report
Comments and Suggestions for Authors
NGS (Next Generation Sequencing) technology enables specialists to pinpoint genes in a short period of time, which greatly guarantees the success rate of IVF. The authors' proposed RCT explores whether the use of PGT-A via NGS is a clinically more efficient, safer, and cost-effective way to provide IVF treatment to advanced reproductive age women. Previous concerns have been clearly explained. The manuscript can be accepted by current form.
Reviewer 3 Report
Comments and Suggestions for Authors
I highly appreciated the extensive effort made bythe authors to reply to all my comments. Unfortunately, the proposed protocol is not convincing as it still lacks the novelty to make significant advance in the field.
I strongly encourage the authors to design a proper RCT study and submit the results to this journal.
Comments on the Quality of English LanguageMinor editing of English language required.